# GRASP: Graph Augmentation via Sampling and Permutation

**Helia Sahebghadam**[1]    Helya.sah@gmail.com
**Ahmed Nebli** [2]    Mr.ahmednebli@gmail.com
[1] *Amirkabir University of Technology*
[2] *Independent Researcher*

**Editors:** Accepted for publication at MIDL 2025

## Abstract

Structural brain graphs illuminate individual differences and neurological traits but are underutilized due to limited data from the challenges of MRI acquisition and preprocessing. We introduce Graph Augmentation via Sampling and Permutation ($GRASP$), a method that synthesizes brain graphs by sampling edge values from consistent positions across multiple adjacency matrices within the same class—assuming topological consistency. Unlike deep learning techniques, $GRASP$ relies on straightforward manipulations of adjacency matrices, which reduces computational demands and simplifies implementation. In this paper, we examine the proof of concept of this augmentation technique on a gender classification task using structural connectomes. We demonstrate enhanced brain graph classification and confirm that within-class adjacency consistency can generate graph variants without complex modeling. The code is publicly available at: https://github.com/heliasah/GRASP-Code

**Keywords:** Structural brain graphs, graph augmentation, topology preservation

## 1. Introduction

Structural connectivity analysis examines anatomical connections between brain regions using Magnetic Resonance Imaging (MRI) (Clayden, 2013). Graph-based methods represent these connections by modeling brain regions as nodes and structural pathways as edges. These models are used to study brain characteristics, including phenotypes such as gender (Kim et al., 2021). For example, (Nebli and Rekik, 2020) modeled brain connectomes as graphs, where nodes represent brain regions and edges encode morphological similarity, then used machine learning to identify gender-specific connections.

Graph-based methods model brain connectivity with high accuracy, but are prone to overfitting when training data is limited (Zhou et al., 2020). This limitation is acute in analyzing brain graphs, where data is acquired via MRI, which requires intensive preprocessing, such as motion correction, alignment, and tissue segmentation. To mitigate this issue, graph augmentation techniques have been introduced, broadly categorized into (I) edge perturbation (e.g., adding, removing, or rewiring edges) (Rong et al., 2020) and (II) node masking (i.e., removing nodes and features) (Feng et al., 2020). Although these methods improve model performance in graph learning tasks, they modify the underlying graph topology by altering node and edge structures. In connectome analysis, such alterations are problematic, as connectivity patterns encode structural attributes. For example, strong connectivity between

the entorhinal cortex and the caudal anterior cingulate cortex has been linked to gender differences (Nebli and Rekik, 2020). Therefore, modifying nodes or edges risks distorting information and might lead to inaccurate diagnosis.

Recent studies, e.g., (Bessadok et al., 2021), have proposed topology-aware approaches like TopoGAN, which preserves connectivity through generative adversarial networks for graph augmentation. However, high training costs and model complexity remain challenges.

In this paper, we assume that new connectivity matrices can be generated by permuting values at corresponding positions across multiple adjacency matrices without disrupting brain graph structure. Based on this, we propose Graph Augmentation via Sampling and Permutation *(GRASP)*, a graph augmentation technique that aims to improve model performance while preserving brain graph topology. Unlike deep learning-based augmentations, our approach is suitable for resource-constrained environments.

## 2. Methods

**Graph Representation:** We model the brain as a weighted directed graph $G = (V, E)$, where $V$ denotes the set of vertices (nodes) and $E$ denotes the set of edges (connections between nodes). Each edge $e \in E$ is assigned a weight by $w : E \to \mathbb{R}$, representing the strength of the connections. For a given graph $G$, the corresponding adjacency matrix $A \in \mathbb{R}^{n \times n}$ is defined, where $n = |V|$ is the number of nodes in the graph. The entry $a_{ij}$ represents the weight of the edge from node $i$ to node $j$, and is given by:

$$A = [a_{ij}] \quad \text{where} \quad a_{ij} = \begin{cases} w(e_{ij}), & \text{if } e_{ij} \in E, \, i, j \in V \\ 0, & \text{if } e_{ij} \notin E, \, i, j \in V. \end{cases} \tag{1}$$

**Assumption 1:** Given a population of brain connectivity matrices $P = \{A^{(1)}, A^{(2)}, \ldots, A^{(m)}\}$, where each $A^{(k)} \in \mathbb{R}^{d \times d}$ represents a graph with $d$ nodes, we define an augmented connectivity matrix $A^{(aug)} \in \mathbb{R}^{d \times d}$ by sampling each entry $A_{i,j}^{(aug)}$ independently from the corresponding entries $\{(A_k)_{i,j}\}_{k=1}^n$. The resulting matrix $A^{(aug)}$ is assumed to preserve the *overall* topological characteristics of the population $P$.

**Augmentation Procedure:** The goal of *GRASP* is to generate new adjacency matrices by sampling from a set of existing ones. Let $\{A^{(1)}, A^{(2)}, \ldots, A^{(m)}\}$ be a set of $m$ adjacency matrices, each of size $n \times n$, corresponding to graphs within the same class label. Each matrix $A^{(k)}$ encodes the connectivity structure of an individual connectome. As shown in Fig. 1, to construct the augmented adjacency matrix $A^{(\text{aug})} \in \mathbb{R}^{n \times n}$, we initialize an empty matrix with the same dimensions as the original adjacency matrices. For each element $a_{ij}^{(\text{aug})}$, we randomly select one of the $m$ available adjacency matrices $A^{(k)}$ from the set $\{A^{(1)}, A^{(2)}, \ldots, A^{(m)}\}$, and assign the corresponding value $a_{ij}^{(k)}$ to the augmented matrix. More formally, for each $i, j \in \{1, 2, \ldots, n\}$, we denote:

$$a_{ij}^{(\text{aug})} = a_{ij}^{(k)} \quad \text{where} \quad k \sim \text{Uniform}(1, m) \tag{2}$$

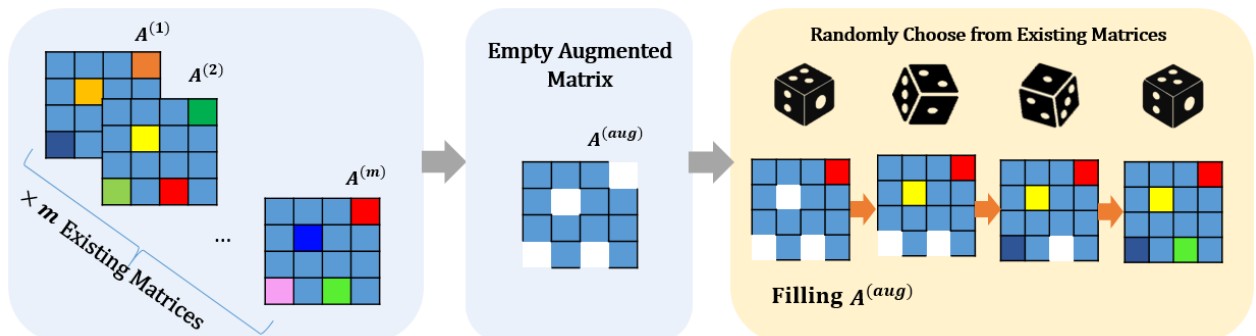

Figure 1: Illustration of the augmentation process. An empty matrix with the dimensions of $A^{(k)}$ is initialized, and each element is filled by randomly choosing from one of the $m$ adjacency matrices

## 3. Results and Discussion

We assess *GRASP* by using the generated graphs for gender classification.

**Dataset:** We used the dataset from (Škoch et al., 2022), which includes 88 subjects (48 females, 40 males) with structural connectomes of 90 brain regions.

**Training Details:** We trained two-layer models (GCN, GAT, GraphSAGE) and a five-layer MLP for 100 epochs on Google Colab, using an 80/20 train-test split. The training set was augmented with 1000 additional adjacency matrices for both male and female brain graphs via *GRASP*. Each classifier was trained on both original and augmented data, and evaluated on the unaugmented test set.

As shown in Table 1, accuracy increased across all models following augmentation, with an average improvement of 20%. The GCN model exhibited the highest gain at 34%, while GAT, GraphSAGE, and MLP also showed less improvements. Despite these gains, the overall increase in accuracy is not *substantial*. We hypothesize that this limitation arises from the random edge selection process used during augmentation, which likely disregards inter-edge dependencies. This is further supported by the performance of the MLP model, which ignores graph structure yet outperformed both GAT and GCN in terms of accuracy.

| Model | Accuracy (Original) | Accuracy (Augmented) |
|---|---|---|
| GCN | 0.52 | 0.70 |
| GAT | 0.59 | 0.64 |
| GraphSage | 0.52 | 0.64 |
| MLP | 0.58 | 0.67 |

Table 1: Classification accuracies for the original and augmented training sets.

## 4. Conclusion

In this paper, we presented *GRASP* which is a graph augmentation technique that constructs matrices by sampling positions from random adjacency matrices, overcoming non-topology-preserving method limitations. It improved connectome gender classification. Future work will refine region-aware sampling to better preserve connectivity patterns.

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
