# OpenReview forum: "GRASP: Graph Augmentation via Sampling and Permutation"
_MIDL.io/2025/Short_Papers — MIDL 2025 - Short Papers_

### Official Review · Reviewer_Qrjq · 2025-04-28

**Rating:** 3
**Confidence:** 5

**Summary:**

This paper proposes a graph augmentation method to address the small sample size problem in brain network analysis. Specifically, the authors introduce a sampling and permutation strategy that synthesizes brain graphs by sampling edge values from multiple adjacency matrices.

**Strengths:**

The paper addresses a critical challenge—the small sample size problem—in brain network analysis. The experimental results are promising, demonstrating that the proposed graph augmentation method can enhance the performance of existing graph neural network (GNN) models. However, the addition of statistical analysis would further strengthen the claims.

**Weaknesses:**

The proposed method does not incorporate topological consistency constraints when generating new graphs, which may limit the structural validity of the augmented samples.

---

### Decision · Program_Chairs · 2025-05-01

Accept